# The Molecular Drivers of Honey Robbing in *Apis mellifera* L.: Morphological Divergence and Oxidative-Immune Regulation Mechanisms Based on Proteomic Analysis

**DOI:** 10.3390/insects16090987

**Published:** 2025-09-22

**Authors:** Xinyu Wang, Xijie Li, Zhanfeng Yan, Mengjuan Hao, Xiao Cui, Zhenxing Liu, Jun Guo, Yazhou Zhao

**Affiliations:** 1Institute of Apicultural Research, Chinese Academy of Agricultural Sciences, Beijing 100093, China; 82101235491@caas.cn (X.W.); 82101235487@caas.cn (M.H.); autumnveslea@163.com (X.C.); liuzhenxing01@caas.cn (Z.L.); 2Faculty of Life Science and Technology, Kunming University of Science and Technology, Kunming 650500, China; lixijie0401@126.com; 3Key Laboratory of Plant Protection Resources and Pest Management of Ministry of Education, College of Plant Protection, Northwest A&F University, Yangling 712100, China; yanmaurice@163.com

**Keywords:** honey robbing, survival analysis, oxidative stress, immunosuppression, proteomics

## Abstract

Honey robbing is an extreme survival strategy used by honey bees during food shortages, but it harms colony health. We studied robber bees and normal foragers, comparing their appearance, lifespan, and head proteins. Key findings: (1) Robber bees have darker bodies due to increased melanin production from higher laccase-5 activity. (2) They live shorter lives because of oxidative stress and weakened immunity—key antioxidant proteins decrease, leading to cell damage, while immune defenses also decline. (3) Their metabolism shifts to boost energy and protein production, helping short-term survival but worsening long-term health. This study provides the first proteomic evidence revealing the physiological trade-offs underlying the behavior at the molecular level, providing new insights into the costs of animal behavior.

## 1. Introduction

Honey bees (*Apis mellifera* L.) as keystone pollinators in ecosystems sustain the reproduction of over 80% of wild plant species globally, and enhance the yield of approximately 65% of crops worldwide [1,2]. Worker bees, the primary labor force of a colony, typically perform nursing duties within 2–3 weeks after emergence before transitioning to foraging activities after 3 weeks. Their foraging efficiency critically determines the colony’s survival: they maintain the colony’s metabolic homeostasis by continuously provisioning nectar and pollen [3,4], while simultaneously establishing a mutualistic “insect-plant” network via pollination services [5,6]. However, under survival pressures such as nectar scarcity or high colony density, some worker bees may violate intraspecific cooperation boundaries and exhibit robbing behavior—invading and robbing food resources from neighboring colonies [7,8] instead of engaging in routine foraging. Although this behavior transiently mitigates resource deficits for individual colonies, it markedly elevates inter-colony pathogen transmission risks. Moreover, it can trigger cascading effects, including queen balling and combat among worker bees. These events propagate rapidly across apiaries, leading to severe consequences such as colony collapse [8,9,10,11].

Floral resource limitation (particularly nectar scarcity) serves as the primary ecological factor triggering the dramatic increase in honey robbing [12]. Current research on honey robbing predominantly focuses on its phenotypic characteristics and ecological drivers. Behavioral analyses reveal that robber bees exhibit an 8-fold increase in aggression and nearly a 2-fold enhancement in extranidal foraging activity compared to control groups [13,14]. Grume et al. observed a significant rise in guard bee numbers at the entrances of robbed colonies, with individuals identifying nestmates through odorant pheromones [14]. Nevertheless, these studies remain largely confined to statistical descriptions of behavior–environment correlations, while scarcely addressing the molecular regulatory pathways underlying the robbing behavior.

In recent years, the application of proteomic technologies in insect behavioral research has made significant progress, providing powerful tools to decipher the molecular mechanisms underlying complex insect behaviors. These techniques have successfully uncovered the proteomic basis of *A. mellifera* caste differentiation [15,16], elucidated the brain protein reprogramming mechanisms during the behavioral transition from nursing to foraging in worker bees [17], and identified olfactory-related protein networks involved in pollen collection decision-making [18]. Furthermore, studies have clarified the molecular regulation of diapause behavior in *Galeruca daurica* [19], characterized energy metabolism features during migratory behavior in *Helicoverpa armigera* [20], and revealed egg-protection mechanisms in the oviposition behavior of *Bombyx mori* Linnaeus [21]. Notably, despite the widespread application of the technology in various insect behavior studies, a significant research gap remains regarding the molecular mechanisms of honey robbing. As an unconventional foraging strategy developed by honey bee colonies under resource competition pressure, the proteomic dynamics and regulatory networks underlying robbing behavior have yet to be systematically investigated.

Addressing this research gap, the present study employs *A. mellifera* as experimental subjects to investigate the physiological costs of behavioral adaptation by comparing morphological traits (including tergite colorimetry, proboscis and appendage dimensions, etc.) and survival rates between robber bees and normal foragers. Furthermore, we implement Data-Independent Acquisition (DIA) mass spectrometry coupled with ultra-high-sensitivity detection to conduct Ultra-Fast quantitative proteomic sequencing of bee head samples. This integrated approach systematically deciphers key metabolic pathways and regulatory nodes associated with behavioral plasticity, ultimately elucidating the molecular proteomic regulatory network underlying the behavior.

## 2. Materials and Methods

### 2.1. Honey Bee Samples

This study employed *A. mellifera* from standardized apiaries at the Institute of Apicultural Research, Chinese Academy of Agricultural Sciences. All colonies used were regularly monitored, with no clinical symptoms of diseases such as *Deformed wing virus*, *Black queen cell virus*, or *Sacbrood virus* detected, and no visual observation of *Varroa destructor* infestation. Seven bee colonies were selected for the experiment, including three strong colonies (gross weight ≥ 40 kg/colony) and four weak colonies (gross weight ≤ 25 kg/colony).

As August marks the peak honey bee robbing season in Beijing, coinciding with a scarcity of nectar and pollen, we marked newly emerged worker bees from the seven colonies in late July for five consecutive days (daily from 07:00 to 10:00). Each day, 300 bees per colony were marked with colored tags and 300 with white tags, with seven distinct colors used to differentiate the colonies. On the 23rd day after the initial marking (in August), the seven colonies were relocated to an open field far from the original apiary. Subsequently, 50% (*w*/*v*) sucrose solution was used as bait, and feeders were placed within one meter in front of the weak colonies’ hives to induce robbing behavior. Researchers captured foreign colored bees (identified as robber bees) inside the robbed hive during real-time monitoring, while simultaneously intercepting white-tagged bees (designated as control bees that did not participate in robbing activities) at the entrances of their original foreign-colored source colony (identified as robbing hive).

### 2.2. Morphometric Measurements

The collected samples were fixed with 75% ethanol and stored at 4 °C for subsequent use. Ten worker bees with no surface damage were randomly selected from both the experimental and control groups as a single sample set. Dissection was performed following Ruttner’s morphological taxonomy method for honey bees [22]: the head, thorax, abdomen, mouthparts, and legs were separated under a stereomicroscope (OLYMPUS SZX16, ×6.3, Olympus Corporation, Tokyo, Japan). Another stereo microscope (ZEISS Stemi 508, ×32, Zeiss Corporation, Oberkochen, Germany) and a digital microscopy imaging system (LEICA DMS 300, equipped with a DFC450 CCD, Leica Microsystems GmbH, Wetzlar, Germany) were used to measure 23 morphological indicators, including proboscis length, width of tomentum on tergite, and appendage dimensions. To minimize operational errors, image acquisition was performed by a designated technician, and standardized analysis was conducted using Digimizer software (v6.02): after importing the original images with scale bars, the software was calibrated to perform three repeated measurements of key parameters (e.g., proboscis length and appendage dimensions), with the mean value taken. All data were exported to Excel 2021 for statistical analysis.

### 2.3. Survival Experiment

The bee collection method was conducted as described in Section 2.1. From both the control bees and the robber bees group samples, 90 worker bees were randomly selected and equally distributed into three rearing cages per group. The custom-designed acrylic cages (150 × 100 × 80 mm^3^) featured precisely arranged 1.5 mm diameter ventilation holes (12 holes/cm^2^ density) on both bottom and side panels, with two integrated 2 mL syringe ports on the upper surface. All experimental groups received continuous provision of 50% sucrose solution (*w*/*v*) under strictly controlled environmental conditions (30 ± 0.5 °C, 60 ± 5% RH) maintained by a calibrated ESPEC PLC-450 environmental chamber. Daily maintenance procedures included 09:00 sucrose solution replacement, survival census recording, and immediate removal of deceased individuals upon verification. The experiment continued until complete natural mortality was achieved across all test groups.

### 2.4. Protein Extraction and Digestion

Protein extraction was conducted following established protocols [18,23]. Head samples from both robber bees and control bees (three worker bee heads were pooled to form one biological replicate, with each group containing four biological replicates, 12 bees) stored at −80 °C were homogenized into powder using liquid nitrogen-precooled mortars. After adding ice-cold lysis buffer (8 M urea, 1 mM PMSF, 2 mM EDTA), samples underwent ice-bath sonication (5 min) followed by centrifugation (15,000× *g*, 10 min, 4 °C; Eppendorf 5425R, Eppendorf, Hamburg, Germany) to obtain supernatants. The total protein concentration was quantified using a BAC assay kit (manufactured by Beyotime Biotechnology, Shanghai, China), with 100 μg protein aliquots subsequently processed through sequential treatments: reduction with 5 mM dithiothreitol (DTT, 37 °C, 30 min) and alkylation with 11 mM iodoacetamide (room temperature, dark, 15 min). Trypsin digestion (sequencing-grade, 37 °C, 16 h) was terminated by acidification to pH 2.0 using 20% trifluoroacetic acid (TFA), followed by C18 column desalting (Millipore, Billerica, MA, USA), vacuum drying, and storage at −20 °C for downstream applications.

### 2.5. LC-MS/MS Analysis

Peptide separation was achieved using a Vanquish Neo UHPLC system with mobile phases consisting of 0.1% (*v*/*v*) formic acid in water (A) and 0.1% (*v*/*v*) formic acid in acetonitrile (B), utilizing a trap-elute configuration with a PepMap Neo Trap Cartridge (300 μm × 5 mm, 5 μm, Thermo Fisher Scientific, Waltham, MA, USA) and an analytical Easy-Spray™ PepMap™ Neo UHPLC column (150 μm × 15 cm, 2 μm, Thermo Fisher Scientific, Waltham, MA, USA) maintained at 55 °C. Following injection of 200 ng sample at 2.5 μL/min, peptides were eluted using a linear gradient from 5% to 25% B over 6.9 min with 8 min column re-equilibration. Eluted peptides were directly ionized via a nano-electrospray source (2.0 kV spray voltage, 275 °C capillary temperature) and analyzed by an Orbitrap Astral mass spectrometer in data-independent acquisition (DIA) mode with positive ion detection. Full MS scans were acquired at 240,000 resolution (at 200 *m*/*z*) across 380–980 *m*/*z* with a 5 ms maximum injection time. MS/MS analysis employed 299 variable isolation windows (2 Th width) using higher-energy collisional dissociation (HCD) at 25% normalized collision energy with an automatic gain control (AGC) target of 500%, 3 ms maximum injection time, and 30 s dynamic exclusion.

### 2.6. Data Analysis

Mass spectrometry raw data acquired from the Orbitrap Astral platform were analyzed using DIA-NN software (v1.8.1) in library-free mode, referencing the NCBI *A. mellifera* proteome database (TaxID: 7460) with decoy database generation (1% false discovery rate threshold) and common contaminant filtering. Protein identification required ≥1 unique peptide with ≥2 matched spectra per protein. Differentially expressed proteins (DEPs) were identified using dual thresholds of |log2(fold change, FC)| ≥ 0.58 (equivalent to FC ≥ 1.5 or ≤0.67) and *t*-test, *p* ≤ 0.05, with differential patterns visualized using R software (v4.5.1) through volcano plots generated with the ggplot2 package (v3.5.0). Functional annotation of DEPs was performed using the Gene Ontology (GO) database (http://geneontology.org/ (accessed on 9 December 2024)), while pathway analysis and enrichment were conducted via the Kyoto Encyclopedia of Genes and Genomes (KEGG) database. Subcellular localization predictions were generated using WoLF PSORT software (v0.2).

### 2.7. Statistical Analysis

Statistical analyses were conducted using SPSS (v27.0). The normality of data for inter-group comparisons was assessed using the Shapiro–Wilk test (*p* > 0.05), confirming that parametric assumptions were met. Subsequently, independent samples *t*-tests were applied for comparisons of morphological characteristics, with a statistical significance threshold set at *p* < 0.05. Survival analyses were performed using the Kaplan–Meier method, and between-group differences were evaluated with the Log-rank test. All visualizations, including survival curves, bar charts, and line graphs, were generated using GraphPad Prism (v9.0). Unsupervised multivariate analyses, including PCA and hierarchical clustering, were carried out using the prcomp function in R software (v4.5.1; www.r-project.org (accessed on 16 December 2024)).

## 3. Results

### 3.1. Differential Morphological Characteristics

This study systematically compared morphological traits between robber bees and control bees (Appendix A), revealing significant differences in only three parameters: tergite 2 pigmentation, tergite 3 pigmentation, and tomentum band width on tergite 4. Specifically, robber bees exhibited darker cuticular coloration, with significantly lower pigmentation values in tergite 2 (Figure 1a) and tergite 3 (Figure 1b) compared to control bees.

The results showed that the width of tomentum on tergite 4 in robber bees was significantly narrower than that in normal foragers (Figure 1c), indicating a reduction in cuticular hair density. This morphological variation may be related to the specific behavioral activities of robber bees, where the decreased cuticular hair density adapts to their high-frequency aggressive fighting activities.

### 3.2. Survival Analysis

The survival curve of control bees displays a typical gradual decline pattern (Figure 2), indicating that honey bees primarily die from natural aging under stable laboratory conditions. In contrast, the survival curve of robber bees shows a significantly accelerated decline rate, with a particularly steep drop after 14 days. Kaplan–Meier survival analysis combined with the Log-Rank test confirmed an extremely significant difference between the two survival curves (*p* < 0.0001). Further calculation of the Hazard Ratio (HR) revealed that robber bees have a 3.576 times higher mortality risk than control bees (95% CI: 2.511–5.092), quantitatively demonstrating the negative effect of robbing behavior on survival.

### 3.3. Identification of Proteins in the Heads

Proteomic sequencing of head proteins from robber bees and control bees identified 8045 unique peptides corresponding to 5218 proteins. PCA showed clear separation between groups with tight clustering of quadruplicate biological replicates (Figure 3a), confirming experimental reliability. Using thresholds of ≥1.5-fold upregulation or ≤0.67-fold downregulation, the *t*-test, and *p* ≤ 0.05, we identified 303 DEPs (184 upregulated, 119 downregulated; Figure 3b, Appendix A). Hierarchical clustering revealed that the DEPs between the two groups displayed distinct category-specific aggregation patterns (Figure 3c). Subcellular localization analysis demonstrated compartment-specific enrichment of DEPs (Figure 3d): the cytoplasm (29.14%) contained metabolic enzyme complexes and signaling molecules; the extracellular matrix (21.52%) was enriched with secretory immune regulators; and the plasma membrane (15.89%) included transmembrane transporters and receptor kinases. The remaining 33.45% predominantly localized to the mitochondria/nucleus, including flavin-containing monooxygenases (FMOs), glucose dehydrogenase (FAD), apyrase, heat shock protein 75 kDa (HSP75), and laccase-5.

### 3.4. GO and KEGG Pathway Analysis of DEPs

To further explore the biological significance of the DEPs, we systematically analyzed the functions of DEPs from three perspectives using the GO annotation system: Molecular Function (MF), Biological Process (BP), and Cellular Component (CC). We statistically evaluated the number of differentially expressed proteins annotated under all secondary GO terms, identifying a total of 29 significant GO terms (Figure 4a). Among these, BP encompassed 14 terms (such as cellular process, metabolic process, localization, and biological regulation), CC included 2 terms (cellular anatomical entity and protein-containing complex), and MF comprised 13 terms (including binding, catalytic activity, structural molecule activity, and transporter activity).

To further investigate the functions of DEPs, we conducted GO enrichment analysis based on the GO annotation results (Figure 4b, Appendix A). In the BP, among the 303 proteins, 47 were involved in translation, 15 in carbohydrate metabolic processes, 3 in actin filament organization, 3 in SRP-dependent cotranslational protein targeting of the membrane, and 2 in the nucleotide catabolic process. For CC, 37 proteins were associated with the ribosome, 6 with the large ribosomal subunit, 23 with the extracellular region, 4 with the cytosolic large ribosomal subunit, 4 with the small ribosomal subunit, and 5 with the extracellular space. In the MF category, 49 proteins were identified as structural constituents of the ribosome, 11 exhibited chitin binding, 6 showed odorant binding, 4 had rRNA binding, 2 possessed nucleotidase activity, and 2 displayed nicotinate phosphoribosyltransferase activity.

To further elucidate the key biochemical metabolic and signal transduction pathways involving DEPs, we conducted an in-depth analysis of the 303 up-/down-regulated proteins using the KEGG database (Figure 4c, Appendix A). The results revealed that these proteins were primarily enriched in two functional categories: protein synthesis/processing and metabolic regulation. The ribosome pathway showed the most significant differential expression, followed closely by protein processing in the endoplasmic reticulum and protein export pathways. These three pathways collectively form a complete “synthesis-processing-export” chain for proteins, all demonstrating significantly activated states in robber bees (with all related DEPs showing upregulation). Among metabolic pathways, notable enrichments were observed in: carbohydrate metabolism (N-glycan biosynthesis, glycosphingolipid biosynthesis, and amino sugar and nucleotide sugar metabolism); nucleotide metabolism (pyrimidine metabolism and nucleotide metabolism); and cofactor metabolism (nicotinate and nicotinamide metabolism and biosynthesis of cofactors). Particularly noteworthy are the pentose phosphate pathway and mTOR signaling pathway—although their enrichment did not reach the statistical significance threshold (Appendix A), they may form a complex cooperative regulatory network through metabolite crossover and signal interaction. These specific metabolic pathway alterations likely reflect the unique metabolic remodeling that occurs in robber bees as an adaptation to their robbing behavior.

## 4. Discussion

### 4.1. Laccase Upregulation Darkens Robber Bees’ Tergite Color

Robber bees exhibit distinct morphological differences compared to control bees, with significantly lower chromaticity values on their pigmentation of tergite 2 and pigmentation of tergite 3 indicating darker body coloration and increased melanin deposition in the cuticle. Laccase, a multi-copper oxidase widely present in insects, not only oxidizes phenolic substances into corresponding quinones [24] but also serves as a crucial sclerotization enzyme [25]. Yamazaki et al. identified a phenol oxidase involved in pupal case hardening and pigmentation in *Drosophila virilis*, later confirmed as laccase [26]. Homologous laccases were also detected in the pupal cuticle of *Drosophila melanogaster* [27]. When Arakane et al. silenced the laccase gene in *Tribolium castaneum* using RNAi, mutants displayed significant phenotypic defects: abnormal cuticle hardening accompanied by pigmentation disorders [28,29]. In our study, the upregulated expression of laccase-5 in robber bees correlated with their altered morphological chromaticity values, suggesting that robber bees may enhance cuticular pigmentation through laccase-mediated sclerotization pathways, resulting in their distinctive body coloration compared to control bees.

### 4.2. Oxidative Stress and Immunosuppression Mediate the Shortened Lifespan of Robber Bees

The study revealed that robber bees exhibit a significantly shortened lifespan, with a mortality risk 3.576 times higher than that of normal foragers (Figure 2). Integrated proteomic analysis provided preliminary insights into the molecular mechanisms underlying their accelerated aging. Firstly, the downregulation of HSP75 suggests impaired mitochondrial protein folding [30], weakening cellular stress response and exacerbating oxidative damage accumulation. Additionally, reduced glutathione transferase (GST) activity indicates compromised clearance of peroxides and electrophilic substances, impairing the antioxidant defense system [31]. Facing this oxidative crisis, robber bees activate a two-tiered defense mechanism: during the primary phase, they upregulate the cytochrome P450 enzyme system (CYP450) and FMOs to enhance xenobiotic metabolism [32,33]; in the secondary phase, peroxiredoxin (Prx) and FAD are upregulated to strengthen antioxidant defenses [34,35]. This compensatory mechanism, however, requires sustained consumption of reducing equivalents such as NADPH. To meet this demand, robber bees activate nicotinate and nicotinamide metabolism to boost NADPH supply. While this metabolic reprogramming temporarily mitigates oxidative stress-induced damage, long-term maintenance of this stress-responsive state not only depletes substantial energy reserves but also disrupts redox homeostasis. Studies have shown [36,37,38] that such imbalance accelerates pro-apoptotic signaling and oxidative damage to biomolecules, thereby hastening the aging process. These findings offer novel molecular-level insights into the aging mechanisms of social insects under environmental stress.

Furthermore, immunity is another critical factor influencing the overall health and survival of honey bees [39,40]. Defense protein 3, an essential effector of innate immunity, plays a vital role in combating various pathogens including bacteria, fungi, and viruses [41,42]. Similarly, C-type lectin 5, a pattern recognition receptor from the C-type lectin family, specifically binds to carbohydrate structures on pathogen surfaces through calcium-dependent recognition mechanisms, playing a pivotal role in immune recognition and clearance [43,44]. This study demonstrates that the reduced expression of these proteins in robber bees impairs their ability to recognize and eliminate pathogens, significantly compromising their health and ultimately shortening their lifespan.

### 4.3. Metabolic and Proteomic Reprogramming in Honey Robbing Adaptation

The high-frequency foraging behavior of robber bees is often accompanied by intense aggressive activities [14], and the sustained execution of such behaviors requires robust support from the organism’s metabolic system. Proteomic analysis revealed that robber bees establish a molecular foundation supporting their behavioral patterns through coordinated upregulation of both energy metabolism and protein synthesis/processing systems. This physiological strategy of “metabolic-synthetic co-enhancement” is primarily manifested in the following aspects.

In terms of energy metabolism, robber bees exhibit significantly enhanced expression of key enzymes involved in nicotinate and nicotinamide metabolism (Nicotinate phosphoribosyltransferase A0A7M7GAK0, A0A7M7R7E2, etc.), nucleotide metabolism (Apyrase A0A7M7GSG4, Adenylate kinase-6 A0A7M7TF44, etc.), and the pentose phosphate pathway (FAD A0A7M7GJ80, A0A7M7RBJ1, etc.). The synergistic effects of these metabolic enzymes form an efficient energy supply network: in the nicotinate and nicotinamide metabolism pathway, nicotinate phosphoribosyltransferase (the key enzyme for NAD+ salvage synthesis) forms a substrate-dependent relationship with FAD from the pentose phosphate pathway, enhancing oxidative phosphorylation efficiency through the NAD+ recycling system [45,46]; in the nucleotide metabolism pathway, the apyrase, adenylate kinase, and protein 5NUC together construct a dynamic energy balance network that both prevents ATP surplus and ensures sufficient emergency energy supply [47,48].

Simultaneously, the protein synthesis system in robber bees demonstrates comprehensive enhancement, manifested by significant upregulation of 51 ribosomal proteins in the ribosome pathway, along with increased expression of Sec61α/β and signal peptidase, collectively establishing an efficient protein folding and secretion system [49]. This enhanced system not only meets the synthetic demands for nutritional functional proteins (such as MRJP and VG) under high metabolic rates [50], but also achieves dual improvements in protein folding quality control and export efficiency through coordinated upregulation of the protein processing in endoplasmic reticulum and protein export pathways. Notably, the activation of the mTOR signaling pathway (with 2 differentially expressed proteins upregulated) reveals the molecular switch mechanism through which robber bees regulate the coordination between energy metabolism and protein synthesis via the nutrient-sensing hub [51,52].

## 5. Conclusions

This study presents the first comparative proteomic analysis between robber bees and normal forager bees. The findings reveal that robber bees establish a molecular foundation supporting their robbing behavior through coordinated upregulation of energy metabolism pathways and protein synthesis/processing systems. However, while this “metabolic-synthetic co-enhancement” adaptation strategy provides short-term behavioral advantages, it comes at the cost of accumulated oxidative damage and downregulation of immune-related proteins, which likely constitute key molecular mechanisms underlying their shortened lifespan. Our results not only uncover the molecular basis of behavioral polymorphism in social insects, but also provide novel molecular evidence for understanding the physiological costs of behavioral adaptation in animals.

While this study provides the first proteomic insights into the molecular mechanisms and physiological costs of robbing behavior adaptation in honey bees, several limitations should be acknowledged. First, all samples were derived from *A. mellifera* colonies in the Beijing region. Future studies could conduct replications across different ecological regions and honey bee species to validate the universality of the relevant molecular mechanisms. Second, although the observed protein expression changes are significantly correlated with behavioral phenotypes, the underlying regulatory mechanisms remain to be thoroughly elucidated. Follow-up research could employ multi-omics approaches (e.g., transcriptomics, metabolomics) to systematically clarify the regulatory network governing the expression of robbing behavior.

## Figures and Tables

**Figure 1 insects-16-00987-f001:**
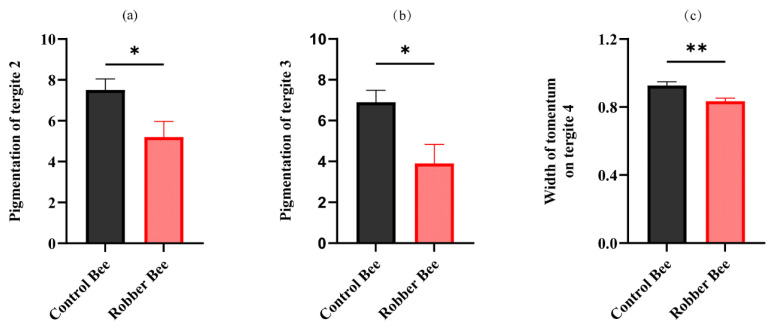
Morphological characteristics between robber bees and control bees. (**a**) Pigmentation of tergite 2 in control bees and robber bees. (**b**) Pigmentation of tergite 3 in control bees and robber bees. (**c**) Width of tomentum on tergite 4 in control bees and robber bees. (data represent the mean ± SD, and statistical analyses were performed using the *t*-test; * represents *p* < 0.05 and ** represents *p* < 0.01).

**Figure 2 insects-16-00987-f002:**
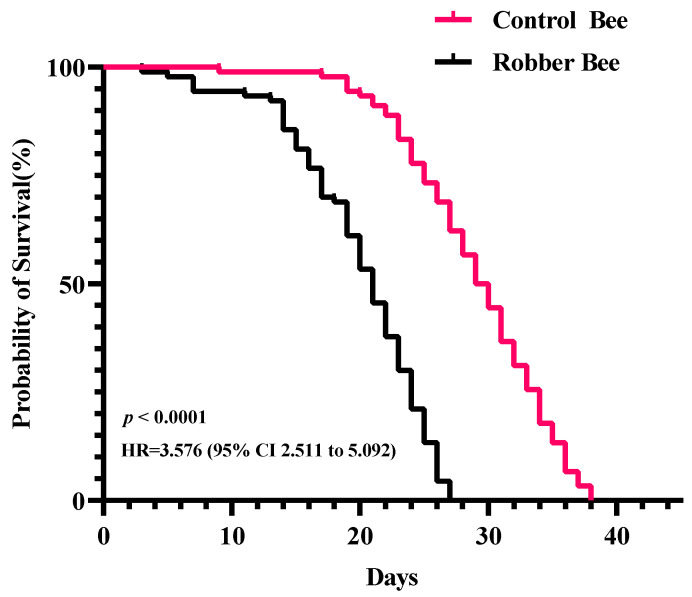
Dynamic comparison of survival rates between robber bees and control bees under laboratory conditions.

**Figure 3 insects-16-00987-f003:**
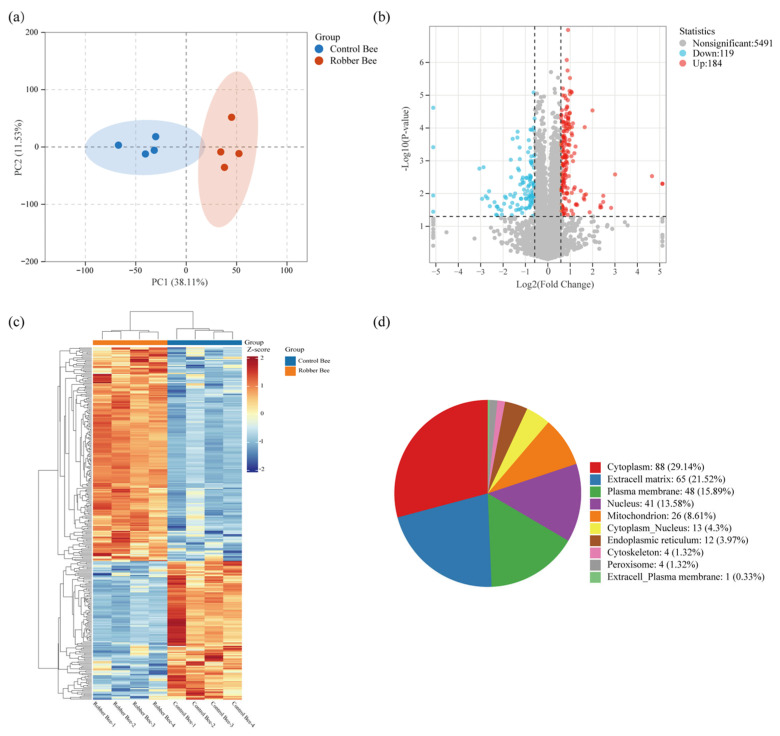
Proteomics Data Analysis (*n* = 4). (**a**) PCA illustrating inter-group variation. (**b**) Volcano plot of DEPs. (**c**) Hierarchical clustering heat-map of DEPs (*Z-score* normalized). (**d**) Subcellular localization of DEPs.

**Figure 4 insects-16-00987-f004:**
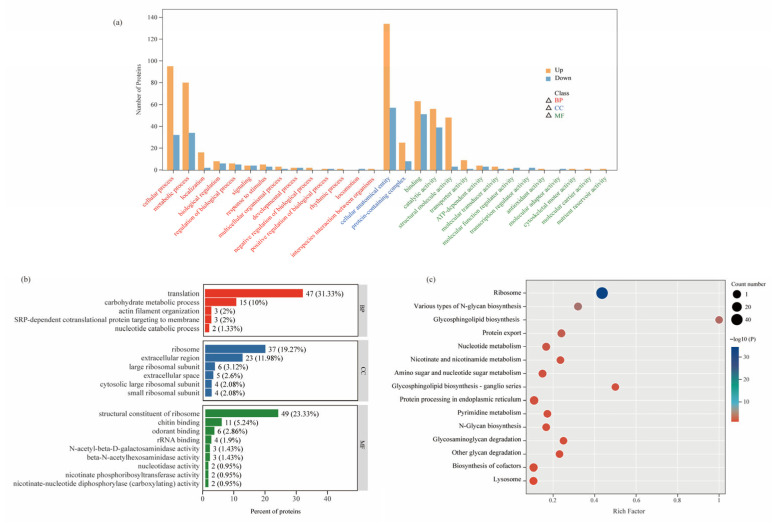
Functional annotation analysis of proteomics. (**a**) GO annotation classification statistics of DEPs. (**b**) GO enrichment analysis of DEPs. The numbers in the figure indicate the count of differentially expressed proteins annotated to the corresponding GO term, with the value in parentheses representing the percentage ratio of the number of differentially expressed proteins annotated to that term relative to the total number of differentially expressed proteins that received functional annotations. (**c**) KEGG enrichment scatter plot of DEPs.

## Data Availability

The raw data supporting the conclusions of this article will be made available by the authors upon request.

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
