# Peer review of "The Molecular Drivers of Honey Robbing in Apis mellifera L.: Morphological Divergence and Oxidative-Immune Regulation Mechanisms Based on Proteomic Analysis"

_insects, 2025, doi:10.3390/insects16090987_

Round 1
Reviewer 1 Report
Comments and Suggestions for Authors
The phenomenon of honeybee robbing can affect the production of beekeeping, and its occurrence mechanism and preventive measures need to be clearly studied.In this study,The authors compared the morphological traits, survival rates and proteins between robber bees and normal foragers, and found significant differences in morphology, longevity, and protein expression. It is very interesting.Comments as belows:
1、The Abstract section should first describe the phenotypic changes in robber bees (including body color and lifespan), followed by the proteomic results, and then provide a combined interpretation of the molecular mechanisms that link the phenotypes to the proteomic findings. By the way, you should tell us that which group of bees are darker and live longer.
2、Line 109, On day 23 post-marking, Why Day 23?So ,the Introduction should include a description of age-related division of labor in honeybees: workers perform nursing and cleaning tasks within 2–3 weeks after emergence, and then transition to foraging activities after 3 weeks. This will provide context for the 23-day post-marking period in the Methods section.
3、After "Apis mellifera" first appears in the main text, the abbreviation "A. mellifera L." or "A. mellifera" should be used instead of the full name; generally, "A. mellifera" is sufficient.
4、Lines 65–66: The citation format should be revised from [8,9,10,11] to [8–11] to comply with the journal’s guidelines.
5、Lines 65–66: The first mention of "Data-Independent Acquisition mass spectrometry" should include the abbreviation in parentheses as "Data-Independent Acquisition (DIA) mass spectrometry".
6、Line 108: Could age variation resulting from marking bees over five consecutive days introduce bias into the results?
7、Line 215 (Figure 2): The line color annotations appear to be incorrect. Does the graph indicate that control bees die faster? Please confirm and revise accordingly.By the way,How did you do this experiment? The bees of day 23 were selected? The experimental method requires further explanation.
8、The discussion is engaging, but the authors should add a paragraph addressing the study’s limitations and future research directions to enhance completeness for readers.
Author Response
Response to Reviewer #1:
Question 1: The Abstract section should first describe the phenotypic changes in robber bees (including body color and lifespan), followed by the proteomic results, and then provide a combined interpretation of the molecular mechanisms that link the phenotypes to the proteomic findings. By the way, you should tell us that which group of bees are darker and live longer.
Response: Thanks for the suggestion from the reviewer. According to the suggestion, we've rechecked and corrected. Changes can be seen in the revised manuscript (Line 33-50).
“The results demonstrated that robber bees exhibited darker tergite coloration and significantly shortened lifespan. Proteomic analysis revealed that the darker coloration was primarily attributed to enhanced cuticular melanin deposition mediated by up-regulated laccase-5, while the shortened lifespan mainly resulted from oxidative stress and immune suppression: the downregulation of heat shock protein 75 kDa and glu-tathione transferase weakened antioxidant capacity, and despite compensatory up-regulation of cytochrome P450 enzyme system, flavin-containing monooxygenases and other enzymes, oxidative damage continued to accumulate. Concurrently, downregulation of Defense protein 3 and C-type lectin 5 caused immune deficiency in robber bees. The results also showed metabolic and protein synthesis reprogramming in robber bees, specifically manifested by upregulated key enzymes in nicotinate and nicotinamide metabolism, pentose phosphate pathway and nucleotide metabolism, along with activation of protein synthesis-transport-export systems. We found that robber bees employ a “metabolic-synthetic co-enhancement” physiological strategy to boost short-term foraging efficiency, but this strategy simultaneously induces oxidative damage and immune suppression, ultimately shortening their lifespan. This study provides the first proteomic evidence revealing the physiological trade-offs underlying this behavior at the molecular level, offering novel insights into the physiological costs of behavioral adaptation in animals”.
Question 2: Line 109, On day 23 post-marking, Why Day 23? So, the Introduction should include a description of age-related division of labor in honeybees: workers perform nursing and cleaning tasks within 2–3 weeks after emergence, and then transition to foraging activities after 3 weeks. This will provide context for the 23-day post-marking period in the Methods section.
Response: Thanks for the reviewer's suggestion. As per the suggestion, we have supplemented the content. The changes can be seen in the revised manuscript. (Line 58-60).
“Worker bees, the primary labor force of a colony, typically perform nursing duties within 2-3 weeks after emergence before transitioning to foraging activities after 3 weeks.”
Question 3: After “Apis mellifera” first appears in the main text, the abbreviation “A. mellifera L.” or “A. mellifera” should be used instead of the full name; generally, “A. mellifera” is sufficient.
Response: Thank you for the reviewer's suggestion. According to the advice, we have rechecked and made corrections accordingly. Use the abbreviation “A. mellifera” uniformly. The specific revisions can be found in the revised manuscript (Line 84, 96, 107, 185, 393).
Question 4: Lines 65–66: The citation format should be revised from [8,9,10,11] to [8–11] to comply with the journal’s guidelines.
Response: Thanks for the suggestion from the reviewer. According to the suggestion, we've rechecked and corrected. Changes can be seen in the revised manuscript (Line 70).
“These events propagate rapidly across apiaries, leading to severe consequences such as colony collapse [8-11].”
Question 5: Lines 65–66: The first mention of “Data-Independent Acquisition mass spectrometry” should include the abbreviation in parentheses as “Data-Independent Acquisition (DIA) mass spectrometry”.
Response: Thanks for the suggestion from the reviewer. According to the suggestion, we've rechecked and corrected. Changes can be seen in the revised manuscript (Line 100).
“Furthermore, we implement Data-Independent Acquisition (DIA) mass spectrometry coupled with ultra-high-sensitivity detection to conduct Ultra-Fast quantitative proteomic sequencing of bee head samples.”
Question 6: Line 108: Could age variation resulting from marking bees over five consecutive days introduce bias into the results?
Response: We thank the reviewer for their suggestion. We would like to clarify that our randomized marking strategy ensured an even distribution of the age variable across both experimental and control groups. Although the marking process spanned five days, by the time sampling was conducted, all marked bees had completed key developmental stages of morphological and physiological maturation within the colonies. At this point, the physiological characteristics of the bees were primarily regulated by behavioral tasks rather than age. Therefore, the slight variation in marking time is unlikely to have introduced bias into the results.
Question 7: Line 215 (Figure 2): The line color annotations appear to be incorrect. Does the graph indicate that control bees die faster? Please confirm and revise accordingly. By the way, How did you do this experiment? The bees of day 23 were selected? The experimental method requires further explanation.
Response: Thanks for the comment of reviewer. Upon verification, it was identified that the color coding of the curves was reversed. The error has now been corrected. Changes can be seen in the revised manuscript (Line 233).
Additionally, although we did not specifically select bees that were exactly 23 days old, all marked bees fell within the same age range. Detailed experimental procedures have been added to the revised manuscript. Changes can be seen in the revised manuscript (Line 142-144).
“The bee collection method was conducted as described in Section 2.1. From both the control bees and the robber bees group samples, 90 worker bees were randomly selected and equally distributed into three rearing cages per group.”
Question 8: The discussion is engaging, but the authors should add a paragraph addressing the study’s limitations and future research directions to enhance completeness for readers.
Response: We would like to thank the reviewer for posing this question. According to the suggestion, we have added to the manuscript. Changes can be seen in the revised manuscript (Line 391-400).
“While this study provides the first proteomic insights into the molecular mechanisms and physiological costs of robbing behavior adaptation in honey bees, several limitations should be acknowledged. First, all samples were derived from A. mellifera colonies in the Beijing region. Future studies could conduct replications across different ecological regions and honey bee species to validate the universality of the relevant molecular mechanisms. Second, although the observed protein expression changes are significantly correlated with behavioral phenotypes, the underlying regulatory mechanisms remain to be thoroughly elucidated. Follow-up research could employ multi-omics approaches (e.g., transcriptomics, metabolomics) to systematically clarify the regulatory network governing the expression of robbing behavior.”

Reviewer 2 Report
Comments and Suggestions for Authors
The study is very interesting, novel and significant for the field, however, lot of methodology date missing (see specific comments bellow), and without them it is impossible to review the significance of the study as well as validity of the hypothesis and conclusions.
Lines 104-105: “with all colonies confirmed pathogen-free before experimentation.” How you tested presence of pathogens? Were they absolutely free of common pathogens such as Varroa, Nosema, viruses etc. or just without clinical signs of diseases? Please add all the details.
Lines 106-107: “the trial involved three strong and four weak colonies relocated to an open field” What was the criteria for classification of colonies as weak or strong? Please add all the details.
Line 110: What is “high-concentration sugar syrup” and what does mean “near weak colonies” Add all details!
Lines 116-117: It is first time you mention control colonies. Were those control colonies some of the listed “three strong and four weak colonies” or some extra colonies?
Lines 130-131: “From both experimental and control group samples, 90 worker bees were randomly selected and equally distributed into three rearing cages per group” Again control groups? How you collect these 90 worker bees, for example from hive inside, brood frames, honey frames, at the entrance?
Within whole manuscript it is not clear what is “control group” as well as “control bee” thorough the text and figures. Are those bees from same haves (but without robber activity) or they are from other hives? Moreover, how you collect them and how you are sure that they are not young bees which did not show robber activity only because of age? Moreover, morphological traits differences could also be due to different age of the bees. This is major problem which need to be clarified and it is very important for relevance of the study.
Line 141: How many samples you had? You mentioned number of biological replicates pre sample but not number of samples per group.
After resolving these issues and providing the above listed details, I will review the paper further (discussion and conclusions).
Author Response
Response to Reviewer #2:
Question 1: Lines 104-105: "with all colonies confirmed pathogen-free before experimentation." How you tested presence of pathogens? Were they absolutely free of common pathogens such as Varroa, Nosema, viruses etc. or just without clinical signs of diseases? Please add all the details.
Response: We thank the reviewer for this question. Prior to the experiment, all colonies showed no clinical symptoms of diseases such as Deformed wing virus, Black queen cell virus, and Sacbrood virus were detected, and no visual signs of Varroa destructor infestation were observed. Infestation by Varroa destructor was assessed using a standard method referenced in the literature: 100 sealed brood cells per colony were randomly selected and visually examined for mites. Only colonies with a mite infestation rate of 0% were included in the study. This information has been added to the revised manuscript as suggested. Changes can be seen in the revised manuscript (Line 108-111).
“All colonies used were regularly monitored, with no clinical symptoms of diseases such as Deformed wing virus, Black queen cell virus, or Sacbrood virus detected, and no visual observation of Varroa destructor infestation.”
Dietemann, V., Nazzi, F., Martin, S. J., Anderson, D. L., Locke, B., Delaplane, K. S. (2013). Standard methods for varroa research. Journal of apicultural research, 52(1), 1-54.
Question 2: Lines 106-107: “the trial involved three strong and four weak colonies relocated to an open field.” What was the criteria for classification of colonies as weak or strong? Please add all the details.
Response: We would like to thank the reviewer for raising this question. As suggested, we have supplemented the manuscript accordingly. We classified the colonies as strong or weak based on their gross weight. Strong colonies were defined as those with a gross weight of no less than 40 kg/colony, while weak colonies were defined as those with a gross weight of no more than 25 kg/colony. The specific revisions can be found in the revised manuscript (Line 111-113).
“Seven bee colonies were selected for the experiment, including three strong colonies (gross weight ≥ 40 kg/colony) and four weak colonies (gross weight ≤ 25 kg/colony).”
Question 3: Line 110: What is "high-concentration sugar syrup" and what does mean “near weak colonies” Add all details!
Response: Thanks for the suggestion from the reviewer. The high-concentration sugar syrup refers to a 50% (w/v)sucrose solution. A feeder containing 50% (w/v) sucrose solution was placed within one meter of weak colonies to induce robbing behavior. Changes can be seen in the revised manuscript (Line 119-121).
“Subsequently, 50% (w/v) sucrose solution was used as bait, and feeders placed within one meter in front of the weak colonies' hives to induce robbing behavior.”
Question 4: Lines 116-117: It is first time you mention control colonies. Were those control colonies some of the listed "three strong and four weak colonies" or some extra colonies?
Response: Thanks for the suggestion from the reviewer. The control group was selected from the aforementioned “three strong and four weak colonies” and was not established as separate, additional colonies. The control group consisted of white-tagged bees from the robbing hive that did not participate in robbing activities.
Question 5: Lines 130-131: “From both experimental and control group samples, 90 worker bees were randomly selected and equally distributed into three rearing cages per group”. Again control groups? How you collect these 90 worker bees, for example from hive inside, brood frames, honey frames, at the entrance?
Response: Thank you for your question. We thank the reviewer for their support and for taking the time to provide detailed constructive feedback on our manuscript. The sampling method was consistent with that described in section 2.1: captured foreign colored bees (identified as robber bees) inside the robbed hive during real-time monitoring, while simultaneously intercepting white-tagged bees (designated as control bees that did not participate in robbing activities) at the entrances of their original foreign-colored source colony(identified as robbing hive). Changes can be seen in the revised manuscript (Line 142-144).
“The bee collection method was conducted as described in Section 2.1. From both the control bees and the robber bees group samples, 90 worker bees were randomly selected and equally distributed into three rearing cages per group.”
Question 6: Within whole manuscript it is not clear what is “control group” as well as “control bee” thorough the text and figures. Are those bees from same haves (but without robber activity) or they are from other hives? Moreover, how you collect them and how you are sure that they are not young bees which did not show robber activity only because of age? Moreover, morphological traits differences could also be due to different age of the bees. This is major problem which need to be clarified and it is very important for relevance of the study.
Response: We thank the reviewer for their suggestion. It should be noted that the control bees were non-robbing individuals from the same hive as the robber bees. The control group consisted of white-tagged bees from the robbing hives that did not participate in robbing activities. Following this suggestion, we have carefully rechecked and revised the manuscript accordingly. The specific modifications can be found in the revised version (Line 121-125).
“Researchers captured foreign colored bees (identified as robber bees) inside the robbed hive during real-time monitoring, while simultaneously intercepting white-tagged bees (designated as control bees that did not participate in robbing activities) at the entrances of their original foreign-colored source colony(identified as robbing hive).”
Additionally, both control bees and robber bees were color-marked as newly emerged adults to eliminate age-related variation. This random marking strategy ensured an even distribution of age variables between the experimental and control groups. Although the marking process spanned five days, by the time sampling was conducted, all marked bees had completed key developmental stages of morphological and physiological maturation within the colonies. At this point, the physiological characteristics of the bees were primarily regulated by behavioral tasks rather than age. Therefore, the slight variation in marking time is unlikely to have introduced bias into the results.
Question 7: Line 141: How many samples you had? You mentioned number of biological replicates per sample but not number of samples per group.
Response: We thank the reviewer for their suggestion. In response, we have now clearly indicated the total sample size for each group in the manuscript. The specific modifications can be found in the revised version (Line 154-157).
“Head samples from both robber bees and control bees (three worker bee heads were pooled to form one biological replicate, with each group containing four biological replicates, 12 bees) stored at -80°C were homogenized into powder using liquid nitrogen-precooled mortars.”

Reviewer 3 Report
Comments and Suggestions for Authors
Honey bee’s robbing activity in resource scarcity season was a typical adaptive strategy, increasing the colony’s competition and survival possibility. However, the background molecular basis was not clear so far. Current study showed that the robber bees have shorter life span, darker tergite coloration and the altered metabolic and protein synthesis activity to boost energy and protein production to meet the requirement their robbing activity. Such result provides a novel insight into the physiological costs and adaption of honeybee robbing activity, thereby is scientifically significant. The following are some suggestions:
- Line 105: “Conducted during Beijing's peak honey robbing season……” , add “bees” after honey.
- Line 107-112: the detail information of honeybee marking and the robber bees collecting should be provided, present information is not enough to understand how the trial was performed, and how the robber bees and control bees were collected. Moreover, the detail age information of the bee samples for LC-MS/MS analysis should be provided too.
- In Survival experiment part, based on the present information, the starting age of both the control and test bees were 23 days old, the survival result showed that the longest survival age could last to 28 and 38 days for control bees and robber bees respectively in Figure 2, which means the life span of test bees could reach to 51 and 61 days for control and robber bees respectively. This longevity data was much longer than most other test. Moreover, the life span of control bees was shorter than the robber bees, which was not consistent with the description of “the survival curve of robber bees shows a significantly accelerated decline rate, with a particularly steep drop after 14 days” in line 208-209. Please check the survival data and description.
- Line 203: name of figure, revise the first capital letter of “Characteristics” into lowercase.
- Line 233: check the format of figure name
Author Response
Response to Reviewer #3:
Question 1: Line 105: “Conducted during Beijing's peak honey robbing season...” , add "bees" after honey.
Response: We thank the reviewer for their suggestion. As recommended, we have supplemented the manuscript accordingly. The corresponding changes can be found in the revised manuscript (Line 114).
“As August marks the peak honey bee robbing season in Beijing, coinciding with a scarcity of nectar and pollen.”
Question 2: Line 107-112: the detail information of honeybee marking and the robber bees collecting should be provided, present information is not enough to understand how the trial was performed, and how the robber bees and control bees were collected. Moreover, the detail age information of the bee samples for LC-MS/MS analysis should be provided too.
Response: We sincerely thank the reviewer for their valuable suggestion, which was extremely helpful to us. Following this recommendation, we have provided a detailed supplement in the revised manuscript regarding the collection methods of both robber bees and control bees. The corresponding modifications can be found in the revised manuscript (Line 114-125).
“As August marks the peak honey bee robbing season in Beijing, coinciding with a scarcity of nectar and pollen, we marked newly emerged worker bees from the seven colonies in late July for five consecutive days (daily from 07:00 to 10:00). Each day, 300 bees per colony were marked with colored tags and 300 with white tags, with seven distinct colors used to differentiate the colonies. On the 23rd day after the initial marking (in August), the seven colonies were relocated to an open field far from the original apiary. Subsequently, 50% (w/v) sucrose solution was used as bait, and feeders placed within one meter in front of the weak colonies' hives to induce robbing behavior. Researchers captured foreign colored bees (identified as robber bees) inside the robbed hive during real-time monitoring, while simultaneously intercepting white-tagged bees (designated as control bees that did not participate in robbing activities) at the entrances of their original foreign-colored source colony(identified as robbing hive).”
The bees used for LC-MS/MS analysis were collected using the aforementioned sampling method and were 18–22 days old.
Question 3: In Survival experiment part, based on the present information, the starting age of both the control and test bees were 23 days old, the survival result showed that the longest survival age could last to 28 and 38 days for control bees and robber bees respectively in Figure 2, which means the life span of test bees could reach to 51 and 61 days for control and robber bees respectively. This longevity data was much longer than most other test. Moreover, the life span of control bees was shorter than the robber bees, which was not consistent with the description of “the survival curve of robber bees shows a significantly accelerated decline rate, with a particularly steep drop after 14 days” in line 208-209. Please check the survival data and description.
Response: We sincerely thank the reviewer for their valuable suggestion, which was extremely helpful to us. Upon verification, it was identified that the color coding of the curves was reversed. The error has now been corrected. The corresponding modifications can be found in the revised manuscript (Line 233).
Regarding the notably longer lifespan observed in this study compared to most similar studies, it should be clarified that the bees used in our survival experiment were not reared in a laboratory setting since their emergence as newly emerged adults. Instead, these bees had completed their entire developmental period within the colonies Throughout the survival assay, both control bees and robber bees were maintained in a laboratory environment. These bees were neither subjected to intensive foraging activities nor exposed to external threats such as predators, pesticides, or harsh weather conditions. These may be factors promoting their longevity.
Question 4: Line 203: name of figure, revise the first capital letter of “Characteristics” into lowercase.
Response: We thank the reviewer for their suggestion. As recommended, we have incorporated corresponding revisions in the revised manuscript. The specific modifications can be found in the revised version (Line 221).
“Figure 1. Morphological characteristics between robber bees and control bees (data represent the mean ± SEM, and statistical analyses were performed using the T-test, * represents P < 0.05, and ** represents P < 0.01).”
Question 5: Line 233: check the format of figure name.
Response: Thank you for your suggestion. We have carefully reviewed this section.

Reviewer 4 Report
Comments and Suggestions for Authors
This is a readable and interesting paper. It needs some revision.
A few minor wording issues are noted on the document.
In the Abstract, line 32, it needs to be made clear that this refers to the robber bees.
In the Data analysis section you should refer to the R software. In the Statistical analysis part, you should state whether the data were normally distributed or if you tested for that before the t-tests. A change to the terminology is also suggested. PCA and cluster analysis are also used and should be mentioned.
Some additional information is needed in the footnote to Table S1 in Additional file 1.
Figure 2 needs corrected- the curves are coloured the wrong way round.
At lines 177, 203/4 and line 222 please mention the test used.
Figure 3(b) needs some correction to the text and the colour coding is confusing – see the annotated document.
All of the Figure 3 and Figure 4 resolution is poor – the text is hard to read. The figures need to be improved.
Some extra detail is needed in the legend of Figure 4(b) – see the annotated document.
The abbreviations and the references need some tidying.
See the annotated document for these and other suggestions.

Author Response
Response to Reviewer #4:
We have compiled all the reviewers' comments from the manuscript.
Question 1: Line 25: un-derlying delete hyphen.
Response: Thanks for the suggestion from the reviewer. According to the suggestion, we've rechecked and corrected the sentence. Changes can be seen in the revised manuscript (Line 26).
“This study provides the first proteomic evidence revealing the physiological trade-offs underlying the behavior at the molecular level, providing new insights into the costs of animal behavior.”
Question 2: Line 25: delete a comma “,”.
Response: Thanks for the suggestion from the reviewer. According to the suggestion, we've rechecked and corrected the sentence. Changes can be seen in the revised manuscript (Line 26).
“This study provides the first proteomic evidence revealing the physiological trade-offs underlying the behavior at the molecular level, providing new insights into the costs of animal behavior.”
Question 3: Line 32: delete “that” but make it clear that this relates to robber bees.
Response: We sincerely appreciate the reviewer's suggestion, which was particularly valuable to us. Following this recommendation, we have supplemented the manuscript accordingly in the revised version. The corresponding modifications can be found in the revised manuscript (Line 33-34).
“The results demonstrated that robber bees exhibited darker tergite coloration and significantly shortened lifespan.”
Question 4: Line 102: above you use “honeybee”. Be consistent-either “honeybee” or “honey bee”.
Response: We thank the reviewer for their suggestion. As recommended, we have consistently used the term "honey bee" throughout the revised manuscript. Changes can be seen in the revised manuscript (Line 18, 29, 56, 93, 106, 114, 130, 226, 340, 392, 395).
Question 5: Line118: thorax, abdomen
Response: We are grateful to the reviewer for their valuable suggestion, which was particularly helpful to us. Following this recommendation, we have revised the manuscript accordingly. Changes can be seen in the revised manuscript (Line 130).
“thorax, abdomen”
Question 6: Line 121: one word or two words?- you use one above.
Response: We thank the reviewer for their suggestion. As recommended, we have used the term "Another " throughout the revised manuscript. Changes can be seen in the revised manuscript (Line 132).
“Another stereo microscope”
Question 7: Line 177: What is the test?
Response: Thank you for your question. The T-test was used in this context. We have supplemented the manuscript accordingly in the revised version. Changes can be seen in the revised manuscript (Line 190).
“Differentially expressed proteins (DEPs) were identified using dual thresholds of | log2(fold change, FC) | ≥ 0.58 (equivalent to FC ≥ 1.5 or ≤ 0.67) and T-test, P ≤ 0.05”.
Question 8: Line 178: refer here to the R software also.
Response: Thank you for your question. The version information of R software has been provided in the revised manuscript. Changes can be seen in the revised manuscript (Line 190-191).
“with differential patterns visualized using R software (v4.5.1) through volcano plots generated with the ggplot2 package (v3.5.0).”
Question 9: Line 183: PCA and cluster analysis should also be mentioned.
Response: Thanks for the suggestion from the reviewer. According to the suggestion, we have added to the manuscript. Changes can be seen in the revised manuscript (Line 204-206).
“Unsupervised multivariate analyses, including PCA and hierarchical clustering, were carried out using the prcomp function in R software (v4.5.1; www.r-project.org).”
Question 10: Line 186: Did you assess the data for normality before using t-tests?
Response: Thanks for the suggestion from the reviewer. According to the suggestion, we have added to the manuscript. Changes can be seen in the revised manuscript (Line 197-201).
“The normality of data for inter-group comparisons was assessed using the Shapiro-Wilk test (P > 0.05), confirming that parametric assumptions were met. Subsequently, independent samples T-tests were applied for comparisons of morphological characteristics, with a statistical significance threshold set at P < 0.05.”
Question 11: Line 188: You mean “bar charts”.
Response: Thank you for your question. We have thoroughly reviewed and corrected this point in the revised manuscript. Changes can be seen in the revised manuscript (Line 203).
“All visualizations, including survival curves, bar charts, and line graphs, were generated using GraphPad Prism (v9.0).”
Question 12: Line 193: refer to Table S1.
Response: Thanks for the suggestion from the reviewer. According to the suggestion, we have added to the manuscript. Changes can be seen in the revised manuscript (Line 209-210).
“This study systematically compared morphological traits between robber bees and control bees (Additional file 1: Table S1).”
Question 13: Line 204: refer here to the statistical test used.
Response: Thanks for the suggestion from the reviewer. According to the suggestion, we have added to the manuscript. Changes can be seen in the revised manuscript (Line 221-223).
“Figure 1. Morphological characteristics between robber bees and control bees (data represent the mean ± SEM, and statistical analyses were performed using the T-test, * represents P < 0.05, and ** represents P < 0.01).”
Question 14: Line 215: the plot is wrongly coloured given what the text says - the black curve here seems to be the one for the robber bee not the control bee
Response: We are grateful to the reviewer for their insightful suggestion, which was particularly valuable. Upon verification, we identified an inadvertent inversion of the curve color labels in Figure 2 and have now corrected this error. Changes can be seen in the revised manuscript (Line 233).
Question 15: Line 222: Which test does this relate to?
Response: Thank you for your question. The T-test was used in this context. We have supplemented the manuscript accordingly in the revised version. Changes can be seen in the revised manuscript (Line 240-242).
“Using thresholds of ≥1.5-fold upregulation or ≤0.67-fold downregulation, T-test, P ≤ 0.05, we identified 303 DEPs (184 upregulated, 119 downregulated; Fig. 3b, Additional file 2: Table S1).”
Question 16: Line 234: the resolution of Figures 3 and 4 needs improvement
Response: We thank the reviewer for their comment. The corresponding vector versions of Figures 3 and 4 have been submitted in the supplementary materials.
Question 17: Line 243: is “such as” needed here?
Response: Thank you for your question. We have removed the relevant content in the revised manuscript. Changes can be seen in the revised manuscript (Line 262-263).
“CC included 2 terms (cellular anatomical entity and protein-containing complex).”
Question 18: Line 278: please state what the numbers are that are shown in the plot (b)
Response: Thank you for your question. We have provided a detailed supplement in the revised manuscript. Changes can be seen in the revised manuscript (Line 297-301). “(b) GO enrichment analysis of DEPs. The numbers in the figure indicate the count of differentially expressed proteins annotated to the corresponding GO term, with the value in parentheses representing the percentage ratio of the number of differentially expressed proteins annotated to that term relative to the total number of differentially expressed proteins that received functional annotations.”
Question 19: Line 369: in the footnote of Table S1 you should mention the test used
Response: Thank you for your question. We have supplemented the relevant information in the updated Additional file 1: Table S1. Changes can be seen in the revised Additional file 1: Table S1.
“Note: data represent the mean ± SEM, and statistical analyses were performed using the T-test, * represents P < 0.05, and ** represents P < 0.01.”
Question 20: Line 388: leave a gap
Response: Thank you for your question. The revisions have been incorporated into the revised manuscript. The changes can be seen in the revised manuscript (Line 421). “Kyoto Encyclopedia of Genes and Genomes, Heat shock protein 75kDa”
Question 21: Line 404: the names in this reference need reformatted
Response: Thank you for your question. The revisions have been incorporated into the revised manuscript. The changes can be seen in the revised manuscript (Line 437).
“7. Ryan Willingham, J.K., and James, E. Robbing Behavior in Honey Bees. EDIS 2021, ENY-163, 163. doi:10.32473/edis-in1064-2015.”
Question 22: Line 406:full stop
Response: Thank you for your question. The revisions have been incorporated into the revised manuscript. The changes can be seen in the revised manuscript (Line 439).
“8. Free, J.B. The behaviour of robber honeybees. Behaviour 1954, 7, 233-240. doi:10.1163/156853955X00085.”
Question 23: Line 412:leave a gap
Response: Thank you for your question. The revisions have been incorporated into the revised manuscript. The changes can be seen in the revised manuscript (Line 445).
“11. Lindström, A.; Korpela, S.; Fries, I. Horizontal transmission of Paenibacillus larvae spores between honey bee (Apis mellifera) colonies through robbing. Apidologie 2008, 39, 515-522. doi:10.1051/apido:2008032.”
Question 24: Line 417:check the name
Response: Thank you for your question. We have reviewed and corrected this issue accordingly. Changes can be seen in the revised manuscript (Line 450).
“14. Grume, G.J.; Biedenbender, S.P.; Rittschof, C.C.; Foraging, Q.; Robbing, S. Honey robbing causes coordinated changes in foraging and nest defence in the honey bee, Apis mellifera. Animal Behaviour 2021, 173, 53-65. doi:10.1016/j.anbehav.2020.12.019.”
Question 25: Line 430: Why is this in red?
Response: Thank you for your question. We have made the corresponding corrections in the revised manuscript. Changes can be seen in the revised manuscript (Line 463).
“19. Ma, H.Y.; Zhou, X.R.; Tan, Y.; Pang, B.P. Proteomic analysis of adult Galeruca daurica (Coleoptera: Chrysomelidae) at I) different stages during summer diapause. Comp Biochem Phys D 2019, 29, 351-357. doi:10.1016/j.cbd.2019.01.007.”
Question 26: Line 439:publisher details?
Response: Thank you for your question. Following the reference format specifications on the official Insects journal website, we have now improved the formatting in the revised manuscript (Line 472).
“22. Ruttner, F. Biogeography and Taxonomy of Honeybees, 1st ed.; Springer: Berlin, Heidelberg, 1988; pp. 66–78.”
Reference example:
Books and Book Chapters:
- Author 1, A.; Author 2, B. Book Title, 3rd ed.; Publisher: Publisher Location, Country, Year; pp. 154–196.
- Author 1, A.; Author 2, B. Title of the chapter. In Book Title, 2nd ed.; Editor 1, A., Editor 2, B., Eds.; Publisher: Publisher Location, Country, Year; Volume 3, pp. 154–196.
Question 27: Line 450: space
Response: Thank you for your question. We have made the corresponding corrections in the revised manuscript. Changes can be seen in the revised manuscript (Line 483).
“27. Barrett, F.M. Phenoloxidases from larval cuticle of the sheepblowfly , Lucilia cuprina: characterization, developmental changes, and inhibition by antiphenoloxidase antibodies. Archives of Insect Biochemistry and Physiology 1987, 5, 99-118. doi:doi.org/10.1002/arch.940050205.”
Question 28: Line 455:use initial capitals for journal name
Response: Thank you for your question. We have made the corresponding corrections in the revised manuscript. Changes can be seen in the revised manuscript (Line 489).
“29. Arakane, Y.; Lomakin, J.; Beeman, R.W.; Muthukrishnan, S.; Gehrke, S.H.; Kanost, M.R.; Kramer, K.J. Molecular and functional analyses of amino acid decarboxylases involved in cuticle tanning in Tribolium castaneum. The Journal of Biological Chemistry 2009, 284, 16584-16594. doi:10.1074/jbc.M901629200.”
Question 29: Line 490: “十”not needed
Response: Thank you for your question. We have reviewed and corrected this issue accordingly. Changes can be seen in the revised manuscript (Line 524).
“42. Ilyasov, R.A.; Gaifullina, L.R.; Saltykova, E.S.; Poskryakov, A.V.; Nikolaenko, A.G. Defensins in the Honeybee Antiinfectious Protection. J Evol Biochem Phys 2013, 49, 1-9. doi:10.1134/S0022093013010015.”
Question 30: Line 495:full stop
Response: Thank you for your question. We have reviewed and corrected this issue accordingly. Changes can be seen in the revised manuscript (Line 528).
“44. Danihlík, J.; Aronstein, K.; Petrivalsky, M. Antimicrobial peptides: a key component of honey bee innate immunity. Physiology, biochemistry, and chemical ecology. J Apicult Res 2015, 54, 123-136. doi:10.1080/00218839.2015.1109919.”

Round 2
Reviewer 2 Report
Comments and Suggestions for Authors
The authors accepted all suggestions and answered all questions and improved the manuscript a lot. Now the manuscript is suitable for publishing.
Author Response
We would like to express our sincere gratitude for your valuable comments and thoughtful guidance on our manuscript. We are pleased to learn that the revised manuscript meets with your approval. All of your suggestions were highly constructive and have significantly enhanced the scientific rigor and clarity of the presentation. Once again, we deeply appreciate the time and effort you dedicated to improving this manuscript. Your insightful feedback has been immensely beneficial to our work.
Reviewer 4 Report
Comments and Suggestions for Authors
I do not seem to have access to the revised Figures 3 and 4 mentioned by the authors, but presume that the authors did improve the resolution and also improved the legend for Fig 3(b) (if not the colour coding).
Table S1 in the Additional File 1 has had additional details of the test added to the footnote, as requested, but the footnote now also refers to the mean +/- SEM, whereas the table legend says “mean and standard deviation”- as also stated in the paper at line 402. The standard deviation and SEM are different. This needs corrected.
Reference 7 still uses full names rather than surnames and initials for the authors.
Author Response
Question 1: I do not seem to have access to the revised Figures 3 and 4 mentioned by the authors, but presume that the authors did improve the resolution and also improved the legend for Fig 3(b) (if not the colour coding).
Response: Thank you for your question. We have made further adjustments to the resolution of Images 3 and 4, achieving the optimal quality achievable for non-vector graphics. (Line 251, 295) Additionally, the revised Figures 3 and 4 have been saved in TIFF format and compiled into a compressed archive named "Figures.zip".
In Figure 3(b), we have replaced the term "insignificant" with "nonsignificant". Since Figure 3(b) compares the upregulation and downregulation of related proteins between control bees and robber bees—rather than representing independent experimental groups—we have accordingly adjusted the color scheme.
Question 2: Table S1 in the Additional File 1 has had additional details of the test added to the footnote, asrequested, but the footnote now also refers to the mean +/- SEM, whereas the table legend says mean and standard deviation- as also stated in the paper at line 402. The standard deviation and SEM are different.
Response: Thank you for pointing this out. We have changed "mean ± SEM" to "mean ± SD" in the footnote of Table S1. The values in the table represent standard deviations and therefore remain unchanged. All values are kept as they were; only the footnote has been unified for consistency. Changes can be seen in the revised Additional file 1: Table S1.
“Note: data represent the mean ± SD, and statistical analyses were performed using the T-test, * represents P < 0.05, and ** represents P < 0.01.”
Question 3: This needs corrected Reference 7 still uses full names rather than surnames and initials for the authors.
Response: Thank you for your question. The revisions have been incorporated into the revised manuscript. The changes can be seen in the revised manuscript (Line 437).
“7. Ryan, W.; Jeanette, K.; and James, E. Robbing Behavior in Honey Bees. EDIS 2021, ENY-163, 163. doi:10.32473/edis-in1064-2015.”
